# MinIE: Minimizing Facts in Open Information Extraction

## Abstract

The goal of open information extraction (OIE) is to extract surface relations and their arguments from natural-language text in an unsupervised, domain-independent manner. In this paper, we explore how overly-specific extractions can be reduced in OIE systems without producing uninformative or inaccurate results. We propose MinIE, an OIE system that produces minimized, annotated extractions. At its heart, MinIE rewrites OIE extractions by (1) identifying and removing parts that are considered overly specific; (2) representing information about polarity, modality, attribution, and quantities with suitable annotations instead of in the actual extraction. We conducted an experimental study with several real-world datasets and found that MinIE achieves competitive or higher precision and recall than most prior systems, while at the same time producing much shorter extractions.

## 1 Introduction

The goal of open information extraction (OIE) is to extract surface relations and their arguments from natural-language text in an unsupervised, domain-independent manner (Banko et al., 2007). In contrast to traditional IE systems, OIE systems do not require an upfront specification of the target schema (e.g., target relations) but instead represent extractions in the form of surface subject-relation-object triples, which are convenient for representing facts in a structured, machine-readable manner (Suchanek et al., 2007; Auer et al., 2007; Bizer et al., 2009). The extractions of OIE systems are useful for tasks such as information retrieval (Löser et al., 2012), question answering (Fader et al., 2014), knowledge-base extension (Dong et al., 2014; Riedel et al., 2013; Petroni et al., 2015) as well as text comprehension, word similarity, word analogy (Stanovsky et al., 2015).

Consider for example the sentence *"Superman was born on Krypton."* An OIE system aims to extract the triple *(Superman, was born on, Krypton)*; in fact, most of the available systems will correctly produce this extraction. A key challenge in OIE is to avoid overly-specific extractions— while simultaneously producing informative and accurate results—when sentences get more complex. To see this, consider the more involved sentence *"Pinocchio believes that the hero Superman was not actually born on beautiful Krypton."* Table 1 shows the corresponding extractions of various OIE systems.[1] Many of the extractions are overly-specific in that the constituents contain specific modifiers or even form complete clauses. In this paper, we explore techniques to minimize OIE extractions and propose a new OIE system called MinIE. Table 1 also shows the output of (two variants of) MinIE for the example sentence.

At its heart, MinIE rewrites OIE extractions by (1) identifying and removing parts that are considered overly specific and (2) removing information about polarity, modality, attribution, and quantities. To retain the original meaning to the extent possible, MinIE provides suitable annotations for each extraction. The idea of using annotations has already been explored by OLLIE (Schmitz et al., 2012), which uses annotations for attribution and clausal modifiers. MinIE follows OLLIE, but uses additional and more expressive annotations. For example, MinIE detects negations in the relation, removes them from the extraction, and adds a "negative polarity" (-) annotation. In fact, MinIE considers the relations *was born on* and *was not born on* equivalent up to polarity. To the best of our knowledge, MinIE is the first OIE system to

---

[1] Suprisingly to us, this particular sentence is problematic for many prior systems. In practice, these systems generally perform better than indicated in this example.

| | | *Pinocchio believes that the hero Superman was not actually born on beautiful Krypton.* | | |
|---|---|---|---|---|
| 1 | OLLIE | (Pinocchio, | believes that, | the hero [...] beautiful Krypton) |
| 2 | | (Superman, | was not actually born on, | beautiful Krypton) |
| 3 | | (Superman, | was not actually born on beautiful Krypton in, | the hero) |
| 4 | ClausIE | (Pinocchio, | believes, | that the hero [...] beautiful Krypton) |
| 5 | | (the hero Superman, | was not born, | on beautiful Krypton) |
| 6 | | (the hero Superman, | was not born, | on beautiful Krypton actually) |
| | Stanford OIE | *No extractions* | | |
| 7 | MinIE-S | (Superman, | was born on, | beautiful Krypton) |
| | | *A.: fact (- [not], CT), attribution (Pinocchio, +, PS [believes]), relation (was actually born on)* | | |
| 8 | | (Superman, | "is", | hero) |
| | | *A.: fact (+, CT)* | | |
| 9 | MinIE-C | (Superman, | was born on, | Krypton) |
| | | *A.: fact (- [not], CT), attribution (Pinocchio, +, PS [believes]), relation (was act. born on), argument (beau. K.)* | | |
| 10 | | (Superman, | "is", | hero) |
| | | *A.: fact (+, CT)* | | |

Table 1: Example extractions and annotations from various OIE systems ("A"=annotation, "+"=positive polarity, "-"=negative polarity, "PS"=possibility, "CT"=certainty)

produce such negative-polarity extractions. The absence of negative evidence is a major concern for reasoning with OIE extractions—e.g., (Dong et al., 2014) proposes to use a "local closed world assumption" and (Riedel et al., 2013; Petroni et al., 2015) use "negative sampling" methods—, and MinIE's annotations help to alleviate this problem.

Apart from polarity annotations, MinIE also provides annotations for modality, attribution, quantities, and for other rewrites that have been performed. Generally, minimizing OIE extractions is inherently limited in scope due to the absence of domain knowledge. Thus MinIE does not and cannot fully minimize its extractions in all cases. Instead, MinIE supports multiple minimization modes, which differ in their aggressiveness. Table 1 shows the output of MinIE's safe mode (which is least aggressive) and its collocation mode (which is more aggressive). We conducted an experimental study with several real-world datasets and found that the various modes of MinIE produced much shorter extractions than prior systems, while simultaneously achieving competitive or higher precision. MinIE sometimes fell behind prior systems in terms of the total number of extractions. We found that those prior system can output a large number of redundant extractions; if redundant extractions were discounted, MinIE was competitive.

## 2 Related work

The concept of OIE was introduced by (Banko et al., 2007). Since then, many different OIE systems have been proposed in the literature. Early systems (Fader et al., 2011) relied on basic NLP techniques—such as POS tagging and chunking—, whereas all recent systems make use of dependency parsing (Gamallo et al., 2012; Wu and Weld, 2010). Most systems represent their extractions in the form of triples, although some systems also produce $n$-ary extractions (Akbik and Löser, 2012; Del Corro and Gemulla, 2013) or use nested representations (Bast and Haussmann, 2013; Bhutani et al., 2016). Some systems focus on noun-mediated extractions (Yahya et al., 2014).

A general challenge in OIE is to avoid both uninformative and overly specific extractions. Re-Verb (Fader et al., 2011) proposed to avoid overly-specific relations by making use of *lexical constraints*: relations that occur infrequently in a large corpus were considered overly specific. MinIE's frequency mode makes use of a similar constraint, but also applies it to subjects and arguments. CSD-IE (Bast and Haussmann, 2013) introduced the notion of nested facts (termed minimal in their paper) and produce extractions with "pointers" to other extractions. NestIE (Bhutani et al., 2016) takes up this idea. MinIE currently does not handle nested extractions.

Perhaps the closest system in spirit to MinIE is Stanford OIE (Angeli et al., 2015), which also uses aggressive minimization. Stanford OIE deletes all subconstituents connected by certain typed dependencies (e.g., *amod*). For some typed dependencies (e.g., *prep* or *dobj*), the system makes use of a frequency constraint along the lines of Re-Verb. MinIE differs from Stanford OIE in that it (1) separates out polarity, modality, attribution, and quantities; (2) uses different, more principled (and more precise) modes of minimization.

The concept of annotated OIE extractions was introduced by OLLIE (Schmitz et al., 2012). OLLIE uses two types of annotations: an *attribution* – describing the the supplier of the triple, and a

*clause modifier* – describing clauses that modify the extraction. MinIE picks up the attribution annotation, and uses additional techniques to provide the attribution's polarity and modality. The clause modifier is an alternative approach to CSD-IE's notion of nested facts and not present in MinIE. MinIE's factuality and quantity annotations have not been provided by existing OIE systems.

## 3 Overview

The goal of MinIE is to provide minimized, annotated OIE extractions. While the techniques employed by MinIE can in principle be integrated into any OIE system, we built MinIE on top of ClausIE (Del Corro and Gemulla, 2013). We chose ClausIE because (1) it separates the identification of the constituents of the extractions from the generation of propositions, (2) it detects clause types, which are also useful for MinIE, and (3) it is a state-of-the-art OIE system.

As ClausIE, MinIE focuses on extractions obtained from individual clauses (with the exception of attribution; see Sec. 5.3). Each clause is broken up into its constituents: one subject (S), one verb (V) and optionally an indirect object (O), a direct object (O), a complement (C) and one or more adverbials (A). ClausIE then identifies the clause type, which indicates which of the constituents are obligatory and which optional from a syntactic point of view. (Quirk et al., 1985) identified the seven clause types SV, SVA, SVC, SVO, SVOO, SVOA, and SVOC; here letters refer to obligatory constituents and each clause can be accompanied by additional optional adverbials.

The general architecture of MinIE is outlined in Fig. 1. Each input sentence is run through ClausIE and a separate extractor for implicit facts (Sec. 4.2). ClausIE's relations are enriched with information from other constituents (Sec. 4.1) in order to make them more informative. We refer to the resulting extractions as *input extractions*. MinIE then rewrites each input extraction to a *basic extraction* by detecting and eliminating information about polarity (Sec. 5.1), modality (Sec. 5.2), attribution (Sec. 5.3), and quantities (Sec. 5.4). To further minimize basic extractions, MinIE can be run in various modes (Sec. 6) with increasing levels of aggressiveness: MinIE-S(afe), MinIE-F(requency), MinIE-C(ollocations), and MinIE-A(ggressive). The modes differ in the amount of minimizations being applied and which additional resources are being used. The result of this phase is a *minimized*

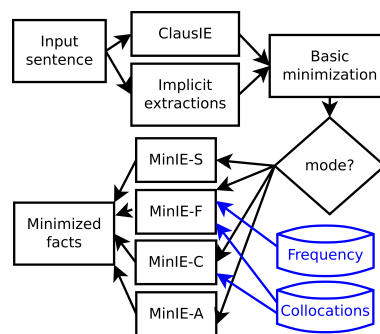

Figure 1: MinIE architecture

*extraction*.

Finally, MinIE outputs each minimized extraction along with annotations that provide information about the minimization process. Some annotations (such as polarity) are crucial to correctly represent the extraction, others (such as original relation) give additional information about which parts have been removed during minimization.

## 4 Input Extractions

Before we discuss how we minimize extractions, we summarize how MinIE obtains its input extractions.

### 4.1 Enriching Relations

As mentioned before, MinIE uses ClausIE as its underlying OIE system. The relations extracted by ClausIE consist of only verbs and negation particles (cf. Tab. 1). (Fader et al., 2011) argues that such an approach can lead to uninformative relations. For example, from the sentence *"Faust made a deal with the Devil"*, ClausIE extracts triple *(Faust, made, a deal with the Devil)*, whereas (Fader et al., 2011) argue that the extraction *(Faust, made a deal with, the Devil)* has a more informative relation. Indeed, the relation *made* is highly polysemous (49 synsets in Word-Net), whereas *made a deal with* is not.

In MinIE, we follow (Fader et al., 2011) and try to produce more informative relations. To do so, we aim to decide which constituents of the input sentence should be pushed into the relation. Our approach is inspired by the syntactic patterns of (Fader et al., 2011), but additionally makes use of the clause type to obtain meaningful relations in a principled way. This allows us, for example, to handle cases with multiple adverbials or with adverbials that do not start with a preposition. Note that the relations produced in this step may sometimes be considered overly specific; they will be minimized further in subsequent steps.

**SVA.** If the adverbial starts with a preposition, we push the preposition into the relation. For example, we rewrite *(Superman, lives, in Metropolis)* to *(Superman, lives in, Metropolis)*. This allows us to distinguish *live in* from relations such as *live during*, *live until*, *live through*, and so on.

**SVOO, SVOC.** We generally push the indirect object (SVOO) or direct object (SVOC) into the relation. In both cases, the verb requires two additional constituents: we use the first one to enrich the relation and the second one as an argument. For example, we rewrite *(Superman, declared, the city safe)* to *(Superman, declared the city, safe)*. As this example indicates, this rewrite is somewhat unsatisfying; further exploration is an interesting direction for future work.

**SVOA.** If the adverbial consists of a single adverb, we push it to the relation and use the object as an argument. Otherwise, we proceed as in SVOC, but additionally push the starting preposition (if present) of the adverbial to the relation. For example, *(Ana, turned, the light off)* becomes *(Ana, turned off, the light)*, and *(The doorman, leads, visitors to their destination)* becomes *(The doorman, leads visitors to, their destination)*.

**Optional adverbials.** If the clause contains optional adverbials, ClausIE creates one extraction per optional adverbial and one without any optional adverbial. We process the former extractions as if the adverbial were obligatory. For example, the extraction *(Faust, made, a deal with the Devil)* becomes *(Faust, made a deal with, the Devil)*. Here the actual clause type is SVO, but we process it as if it were SVOA.

**Infinitive forms.** If the argument starts with a to-infinitive verb, we move it to the relation. For example, *(Superman, needs, to defeat Lex)* becomes *(Superman, needs to defeat, Lex)*.

### 4.2 Implicit Extractions

ClausIE produces non-verb-mediated extractions from appositions and possesives. We refer to these extractions as *implicit extractions*. In MinIE, we use implicit extractions as a signal for the minimization of non-implicit ones (see Sec. 6.1). To make the minimization more effective, we include additional patterns identified in prior work. We use the explicit patterns of FINET (Del Corro et al., 2015) to detect explicit type mentions. For example, if the sentence contains *president Barack Obama*, we obtain *(Barack Obama, is, president)*. We also include certain patterns involving named entities: pattern *ORG IN LOC* for extraction *(ORG, is IN, LOC)*; pattern *"Mr." PER* for

| Sentence | Factuality |
|---|---|
| S. does live in Metropolis. | (+, CT) |
| S. does not live in M. | (– [not], CT) |
| S. does probably live in M. | (+, PS [probably]) |
| S. probably does not live in M. | (– [not], PS [probably]) |

Table 2: Factuality examples. MinIE extracts triple *(Superman; does live in; Metropolis)* from each sentence but the factuality annotations differ.

*(PER, is, male)* (similarly, Ms. or Mrs.); and pattern *ORG POS? NP PER* for *(PER, is NP of, ORG)* from RelNoun (Pal and Mausam, 2016).

## 5 Basic Minimization

The basic minimization step of MinIE detects and eliminates information about polarity (Sec. 5.1), modality (Sec. 5.2), attribution (Sec. 5.3), and quantities (Sec. 5.4). Our focus is on methods that are both domain-independent and (considered) safe to use.

### 5.1 Polarity

MinIE annotates each extraction with information about its *factuality*. Following (Saurí and Pustejovsky, 2012), we represent the factuality of an extraction with two pieces of information: polarity (+ or -) and modality (CT or PS; for certainty or possibility, resp.). Tab. 2 lists some examples.

The *polarity* indicates whether or not a triple occured in negated form. In order to assign a polarity value to a triple, we aim to detect whether the relation indicates a negative polarity. If so, we assign negative polarity to the whole triple. We detect negations using a small lexicon of negation words (consisting of the words *no, not, never, none*). Whenever a word from the lexicon is detected, it is dropped from the relation and the triple is annotated with negative polarity (-) and the negation word. In Tab. 2, the extractions from sentences 2 and 4 are annotated as negative.

We found that this simple approach successfully spots many negations present in the input relations. Note that whenever a negation is missed, MinIE still produces correct results because such negations are retained in the triple. For example, if a negations occurs in the subject or argument of the extraction, MinIE misses it. E.g., from sentence *"No people were hurt in the fire"*, MinIE extracts *(Q₁ people, were hurt in; fire)* with quantity $Q_1 = no$ (see Sec. 5.4). This extraction is correct, but can be further minimized to *(people; were hurt in; fire)* with a negative polarity annotation. We consider such advanced minimizations too danger-

ous to use in the absence of domain knowledge.

Generally, negation detection is a hard problem and involves questions such as negation scope resolution, focus detection, and double negation (Blanco and Moldovan, 2011). MinIE does not address these problems, but restricts attention to the easy cases.

## 5.2 Modality

The *modality* of a triple indicates whether the triple is considered a *certainty* (CT) or a *possibility* (PS) in the clause in which it occurs. We proceed similarly to the detection of negations and consider a triple certain unless we find evidence indicating possibility.

To find such evidence, MinIE searches the relation for (1) modal verbs such as *may* or *can*, (2) possibility-indicating words, and (3) certain infinitive verb phrases. For (2) and (3), we make use of a small domain-indepedent lexicon. Our lexicon is based on the lexicon of (Saurí and Pustejovsky, 2012) and the words in the corresponding WordNet synsets. It mainly contains adverbs such as *probably, possibly, maybe, likely* and infinitive verb phrases such as *is going to*, *is planning to*, or *intends to*. Whenever words indicating possibility are detected, we remove these words from the triple and annotate the triple as possible (PS) along with the words just removed. For example, sentences 3 and 4 in Tab. 2 are annotated PS with the possibility-indicating word *probably*.

As with negation detection, MinIE's approach focuses on the easy cases and uses only the CT and PS levels. A more advanced approach is taken by (Saurí and Pustejovsky, 2012).

## 5.3 Attribution

The *attribution* of a triple is the supplier of information given in the input sentence, if any. We adapt our attribution annotation from the notion of *source* in (Saurí and Pustejovsky, 2012), i.e., the attribution consists of a supplier of information along with a factuality (polarity and modality). The factuality is independent from the factuality of the extracted triple; it indicates whether the supplier expresses a negation or a possibility. Tab. 1 shows some examples.

We extract attributions from subordinate clauses as well as from an "according to" pattern. For the former, MinIE searches for extractions that contain entire clauses as arguments. We then compare the relation against a domain-independent dictionary of relations indicating attributions (e.g., *say*

or *believe*). If we find a match, we create an attribution annotation and use the subject of the extraction as the supplier of information. Each entry in the attribution dictionary is annotated with a modality. For example, relations such as *know*, *say*, or *write* express certainty, whereas relations such as *believe* or *doubt* express possibility. If the relation is modified by a negation word (see Sec. 5.1), we mark the attribution with negative polarity (e.g., *never said that*). After the attribution has been established, we run ClausIE on the main clause and add the attribution to each extracted triple. The second source of attributions is obtained via "according to" adverbial phrases. To detect those, we search for adverbials that start with *according to* and take whatever follows as the supplier with factuality *(+,CT)*. The remaining part of the clause is processed as before.

## 5.4 Quantities

A *quantity* is a phrase that expresses an amount of something. It either modifies a noun phrase (e.g. *9 cats*) or is an independent complement (e.g. *I have 3*). Quantities include cardinals (*9*), determiners (*all*) or whole phrases (*almost 10*). Whenever we detect a quantity, we replace it by a placeholder $Q$ and add an annotation with the original quantity. The goal of this step is to unify extractions that only differ in quantities. For example, the arguments *9 cats*, *ten cats*, *all the cats* and *almost about 100 cats* are all rewritten to *Q cats*, only the quantity annotation differs.

We detect quantities by looking for numbers (NER types such as NUMBER or PERCENT) or words expressing quantities (such as *all, some, many*). We then extend these words via relevant typed dependencies, such as quantity modifiers (*quantmod*) and or adverbial modifiers (*advmod*).

## 6 Minimization Modes

Once basic minimization has been performed, MinIE further minimizes extractions by dropping additional words. Since such minimization is risky, MinIE employs various minimization modes with different levels of aggressiveness. We discuss these modes in turn, from the safest mode to the most aggressive one.

MinIE represents each constituent of a basic extraction by its words, its dependency structure, its part-of-speech tags, and its named entities (detected by a named-entity recognizer). In general, each mode defines a set of *stable subconstituents*, which will always be fully retained, and subse-

quently searches for candidate words to drop outside of the stable subconstituents. Whenever a word is dropped from a constituent, we add an annotation with the original, unmodified constituent. In all of MinIE's modes, named entites are considered stable subconstituents.

## 6.1 Safe Mode (MinIE-S)

MinIE's *safe mode* only drops word which we consider universally safe to drop. We first drop all constituents that are covered by the implicit extractions discussed in Sec. 4.2; e.g., types of named entities, locations of organizations, and the words "Mr.", "Ms.", and "Mrs." before persons. We then drop all determiners, all possessive pronouns, all adverbs modifying the verb in the relation, and all adjectives and adverbials modifying persons. An exception to these rules is given by named entities, which we consider as stable subconstituents.

Note that this procedure cannot be considered safe when used on input extractions. We consider it safe, however, when applied to basic extractions. In particular, all determiners, pronouns, and adverbs indicating negation, modality, or quantities are already processed during basic minimization and are consequently retained in annotations. Moreover, due to the way we construct input extractions, each relation has at most one adverb modifying its verb. The safe mode thus only performs simple rewrites such as *the great city* to *great city*, *his presidency* to *presidency*, *had also* to *had*, and *the eloquent president Mr. Barack Obama* to *Barack Obama*.

## 6.2 Frequency Mode (MinIE-F) and Collocations Mode (MinIE-C)

The frequency and collocations modes make use of a *dictionary $\mathcal{D}$ of stable constituents*. The two modes differ only in the choice of $\mathcal{D}$. We first discuss how the dictionary is being used, and subsequently how we can construct it.

Both modes first perform all the minimizations of the safe mode and then search for maximal noun phrases of the form $P \equiv$ *[adverbial|adjective]$^+$ [noun$^+$|ner]*. For each instance of $P$, we drop a certain subset of its words. For example, a suitable minimization for *very infamous cold war symbol* (i.e., the Berlin wall) is *cold war symbol*, i.e., we consider *cold* as essential to the meaning of the constituent and *very infamous* as overly specific. The decision of what is considered essential and what overly specific is informed by dictionary $\mathcal{D}$. Note that in order to minimize mistakes, we con-

sider for dropping only words in instances of pattern $P$. In particular, we do not touch subconstituents that contain prepositions because these are notoriously difficult to handle (e.g., we do not want to minimize *Bill of Rights* to *Bill*).

Our main goal is to retain phrases that occur in $\mathcal{D}$, even if they occur in different order or with additional modifiers. To achieve this goal, we proceed as follows for each instance $I$ of $P$. We first mark all nouns (or the named entity) as *stable*. Afterwards, we create a set of *potentially stable subconstituents* (PSS). Each PSS is queried against dictionary $\mathcal{D}$. If it occurs in $\mathcal{D}$, all of its words are marked as stable. Once all PSS have been processed, we drop all words from $I$ that are not marked stable. In our example, if $\mathcal{D} = \{cold\ war\}$, we obtain the minimization *cold war symbol* as desired.

To generate the set of PSS, we enumerate all subconstituents of $I$ that are syntactically valid. For example, *infamous symbol* or *cold infamous war* are syntactically valid, whereas *very symbol* or *very cold war* are not. Conceptually,[2] we enumerate all subsequences of $I$ and check whether (1) at least one noun (or the named entity) is retained, and (2) whenever an adverbial or adjective has been dropped, so are all of its modifiers. For each such subsequence, we generate all permutations of adverbial and adjective modifiers originating from the same node in the dependency structure. This step ensures that the order of modifiers in $I$ does influence whether or not a word is marked stable. The set of PSS for *very infamous cold war symbol* contains 22 entries.

It remains to discuss the construction of dictionary $\mathcal{D}$. In MinIE's collocations mode, $\mathcal{D}$ is provided externally and contains a set of collocations. In our experimental study, we included all multi-word expressions from WordNet and Wiktionary, which we consider domain-independent. In practice, applications can extend the dictionary using suitable collocations, either from domain-dependent dictionaries or by using methods to automatically extract collocations from a text corpus (Gries, 2013). In MinIE's frequency mode, we process the entire corpus using the safe mode and additionally include all frequent (e.g., frequency $\geq 10$) subjects, relations, and arguments into $\mathcal{D}$. This *lexical constraint* is inspired by (Fader et al., 2011); the rationale is that everything sufficiently frequent is not considered overly specific.

---

[2]In our implementation, we generate both instances of $P$ as well as the set of PSS directly from the dependency structure of the constituent.

## 6.3 Aggressive Mode (MinIE-A)

The modes discussed so far aimed to be conservative. MinIE-A proceeds the other way around: all words for which we are not sure that they need to be retained are dropped. We mainly include this mode as a baseline and to study how effective such an approach is empirically.

For every word in a constituent of a basic extraction, we drop all adverbial, adjective, possessive, and temporal modifiers (recursively along with their modifiers). We also drop prepositional attachments (e.g., *man with apples* becomes *man*), quantifiers modifying nouns, auxiliary modifiers to the main verb (e.g., *have escalated* becomes *escalated*), and all compound nouns that have a different named-entity type than their head word (e.g., *European Union offical* becomes *offical*). In most cases, after applying these steps, only a single word, named entity, or a sequence of nouns remains for subject and argument constituents.

## 7 Experimental Study

The goal of our experimental study was to investigate the differences in the various modes of MinIE w.r.t. precision, recall, and extraction length as well as to compare it to popular prior methods.

### 7.1 Experimental Setup

All source code, datasets, extractions, labels, and labelling guidelines will be made publically available. We also submitted them to SoftConf.

**Datasets.** We used as datasets (1) a random sample of 10,000 sentences from the New York Times Annotated Corpus (NYT-10k), (2) a random sample of 200 sentences from the same corpus (NYT), and (3) a random sample of 200 sentences from Wikipedia (Wiki). Datasets NYT and Wiki were used in the evaluation of ClausIE and NestIE.

**Methods.** We used ClausIE, OLLIE, and Stanford OIE as baseline systems. We adapted the publically available version of ClausIE to Stanford CoreNLP 3.5.1 (Manning et al., 2014) and implemented MinIE on top. For MinIE-F, we built the dictionary using the entire NYT corpus.

**Labeling.** We labeled every extraction of NYT and Wiki with two labels: one for the triple and one for the attribution. A triple is labeled as correct if it is entailed by its corresponding clause; here factuality annotations are taken into account. For example, all triples except #3 of Tab. 1 are considered correct. The attribution is labeled incorrect if there is an attribution which is neither present in the triple nor in the attribution annotation. In Tab. 1, the attribution is incorrect for extractions 2, 3, 5, and 6. We only label attribution when the fact triple is labeled correct.[3] Further details can be found in the labelling guidelines.

**Measures.** For each system, we measured the total number of extractions (termed *recall*), the fraction of correct triples (*factual precision*), and the fraction of correct triples with correct attributions (*attribution precision*). We also determined the mean word count per extraction ($\mu$) and its standard deviation ($\sigma$); we use these numbers as a proxy for minimality. Finally, we observed that some systems produced a large number of redundant extractions. To obtain insight into the amount of redundancy, we also report the number of non-redundant extractions. For simplicity, we consider a triple $t_1$ redundant if it appears as subsequence in some other triple $t_2$ produced by the same extractor from the same sentence (e.g., extraction 5 in Tab. 1 is redundant given extraction 6).

### 7.2 Extraction Statistics

In our first experiment, we used the larger NYT-10k dataset (which we did not label). Tab. 3 summarizes statistics of the extraction produced by each system. For MinIE, we show the fraction of negative polarity and possibility annotations for triples only (i.e., we exclude the attribution annotation).

In terms of recall, MinIE and Stanford OIE were roughly on par; OLLIE fell behind and ClausIE went ahead. The reason why ClausIE has more extractions than MinIE is that different (partly redundant) extractions from ClausIE may lead to the same minimized extraction. This is also also the reason why recall drops in the more aggressive modes of MinIE. We also determined the number of non-redundant extractions produced by each system and found that most system's only produce a moderate number of redundant extractions. A notable exception is Stanford OIE, which produces many extraction variants by dropping different subsets of words.

We observed that all modes of MinIE achieved significantly smaller extractions than ClausIE (its underlying OIE system), and that the average extraction length indeed dropped as we used more aggressive modes. Only MinIE-A produced shorter extractions than Stanford OIE. The main reason for the short extraction length of Stanford

---

[3]In fact, it's meaningless to attribute incorrect extractions to suppliers.

|                       | OLLIE           | ClausIE       | Stanford      | MinIE-S       | MinIE-F       | MinIE-C       | MinIE-A         |
|-----------------------|-----------------|---------------|---------------|---------------|---------------|---------------|-----------------|
| # non-redundant extr. | 21,277          | **41,174**    | 16,779        | 37,638        | 37,698        | 37,277        | 36,155          |
| # extractions         | 24,316          | **58,420**    | 43,365        | 45,649        | 45,386        | 45,322        | 43,035          |
| $\mu \pm \sigma$      | $9.3 \pm 5.2$   | $10.9 \pm 7$  | $6.6 \pm 3.0$ | $7.2 \pm 4.2$ | $6.9 \pm 4.0$ | $6.8 \pm 4.0$ | **$4.7 \pm 1.9$** |
| with attributions     | 6.8%            | -             | -             | 10.7%         | 10.69%        | 10.7%         | 10.8%           |
| with negative polarity| -               | -             | -             | 3.7%          | 3.7%          | 3.7%          | 3.8%            |
| with possibility      | -               | -             | -             | 9.9%          | 9.9%          | 9.9%          | 9.7%            |
| with quantity         | -               | -             | -             | 17.8%         | 17.8%         | 17.8%         | 1.9%            |

Table 3: Results on the unlabeled NYT-10k dataset ($\mu$=avg. extraction length, $\sigma$=standard deviation)

|      |                 | OLLIE    | ClausIE      | Stanford     | MinIE-S     | MinIE-F  | MinIE-C  | MinIE-A  |
|------|-----------------|----------|--------------|--------------|-------------|----------|----------|----------|
| NYT  | # non-redundant | 306/435  | 621/**928**  | 218/364      | **623**/791 | 618/792  | 609/785  | 514/748  |
|      | # extractions   | 353/497  | **890/1300** | 649/1052     | 730/923     | 717/914  | 713/914  | 588/861  |
|      | factual prec.   | (0.71)   | (0.68)       | (0.62)       | **(0.79)**  | (0.78)   | (0.78)   | (0.68)   |
|      | attr. prec.     | (0.92)   | -            | -            | **(0.94)**  | **(0.94)** | (0.93)  | (0.93)   |
| Wiki | # non-redundant | 343/491  | **561/777**  | 261/431      | 558/676     | -        | 540/674  | 459/663  |
|      | # extractions   | 399/565  | 728/1002     | **812/1516** | 683/822     | -        | 653/819  | 528/788  |
|      | factual prec.   | (0.71)   | (0.73)       | (0.54)       | **(0.83)**  | -        | (0.80)   | (0.67)   |
|      | attr. prec.     | **(0.98)** | -          | -            | **(0.98)**  | -        | **(0.98)** | **(0.98)** |

Table 4: Results on the labeled NYT and Wiki datasets

OIE is its aggressive creation of (shorter) redundant extractions.

Only OLLIE and MinIE make use of annotations. The fraction of extracted attribution annotations was significantly smaller for OLLIE than for MinIE, mainly because OLLIE's attribution detection is limited to the *ccomp* dependency relation. The statistics also indicate that the basic minimizations of MinIE fire quite frequently, with the notable exception of negations.

### 7.3 Precision

In our second experiment, we compared the precision of the various systems on the smaller NYT and Wiki datasets. We did not run MinIE-F on Wiki because it was trained on the full NYT corpus. Our results are summarized in Tab. 4, where we give recall in the form of #correct/#total.

We found that Stanford OIE had the lowest factual precision throughout, producing many incorrect and many redundant extractions (e.g., Stanford OIE produced 400 extractions from five sentences on NYT). For MinIE, the factual precision dropped as expected when we use more aggressive modes. Interestingly, the drop in precision between MinIE-S, MinIE-F, and MinIE-C was quite low, even though extractions get shorter. The aggressive minimization of MinIE-A led to a more severe drop in precision. Surprisingly to us, even MinIE's aggressive mode achieved higher precision than Stanford OIE. ClausIE and OLLIE had similar factual precision. Note that MinIE-S had higher precision than ClausIE. Reasons include that MinIE-S produces additional high-precision implicit extractions, breaks up very long and thus

error-prone extractions, and uses a different version of Stanford CoreNLP.

As for attribution precision, most of the sentences in our samples did not contain attributions; these numbers should thus be taken with a grain of salt. OLLIE and MinIE achieved similar results, even though MinIE additionally annotated attributions with factuality information.

### 7.4 MinIE's Errors

For all modes, errors in dependency parsing transfer over to errors in MinIE, which we believe is the main source of errors in MinIE-S. For MinIE-C and MinIE-F, we sometimes drop adjectives which in fact form collocations (e.g., "*assistant* director") with the noun they are modifying, but the collocation is not present in the dictionary and thus broken up. This problem can be addressed by using richer collocation dictionaries. Another source of error stems from the named-entity recognizer (e.g., the first word of the named entity *Personal Ensign* was not recognized so that MinIE incorrectly extracts *("Ensign", "is", "Personal")*).

## 8 Conclusions

We presented MinIE, a system that minimizes and annotates OIE extractions. We believe that the use of minimized extractions with suitable annotations are a promising direction for OIE. The techniques presented here can be seen as a step towards this goal. Future directions include additional annotation types (e.g. temporal/spatial), improved techniques for basic minimizations, better handling of collocations, and use of nested representations.

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
