# Peer review of "MinIE: Minimizing Facts in Open Information Extraction"

_ACL 2017 — decision unknown_

[Official Review · Reviewer 1 · rating 3 · confidence 4]
soundness 3 · originality 4 · clarity 4 · impact 3 · substance 3 · appropriateness 5 · meaningful comparison 3 · presentation format Poster

- Strengths:

[+] Well motivated, tackles an interesting problem;

[+] Clearly written and structured, accompanied by documented code and dataset;

[+] Encouraging results.

- Weaknesses:

[-] Limited to completely deterministic, hand-engineered minimization rules;

[-] Some relevant literature on OIE neglected;

[-] Sound but not thorough experimental evaluation.

- General Discussion:

This paper tackles a practical issue of most OIE systems, i.e. redundant,
uninformative and inaccurate extractions. The proposed approach, dubbed MinOIE,
is designed to actually "minimize" extractions by removing overly specific
portions and turning them into structured annotations of various types
(similarly to OLLIE). The authors put MinIE on top of a state-of-the-art OIE
system (ClausIE) and test it on two publicly available datasets, showing that
it effectively leads to more concise extractions compared to standard OIE
approaches, while at the same time retaining accuracy.

Overall, this work focuses on an interesting (and perhaps underinvestigated)
aspect of OIE in a sound and principled way. The paper is clearly written,
sufficiently detailed, and accompanied by supplementary material and a neat
Java implementation.
My main concern is, however, with the entirely static, deterministic and
rule-based structure of MinIE. Even though I understand that a handful of
manually engineered rules is technically the best strategy when precision is
key, these approaches are typically very hard to scale, e.g. in terms of
languages (a recent trend of OIE, see Faruqui and Kumar, 2015; Falke et al.,
2016). In other words, I think that this contribution somehow falls short of
novelty and substance in proposing a pipeline of engineered rules that are
mostly inspired by other OIE systems (such as ClausIE or ReVerb); for instance,
I would have really appreciated an attempt to learn these minimization rules
instead of hard-coding them.

Furthermore, the authors completely ignore a recent research thread on
“semantically-informed” OIE (Nakashole et al., 2012; Moro and Navigli,
2012; 2013; Delli Bovi et al., 2015) where traditional extractions are
augmented with links to underlying knowledge bases and sense inventories
(Wikipedia, Wikidata, Yago, BabelNet). These contributions are not only
relevant in terms of related literature: in fact, having text fragments (or
constituents) explicitly linked to a knowledge base would reduce the need for
ad-hoc minimization rules such as those in Sections 6.1 and 6.2. In the example
with "Bill of Rights" provided by the authors (line 554), an OIE pipeline with
a proper Entity Linking module would recognize automatically the phrase as
mention of a registered entity, regardless of the shape of its subconstituents.
Also, an underlying sense inventory would seamlessly incorporate the external
information about collocations and multi-word expressions used in Section 6.2:
not by chance, the authors rely on WordNet and Wiktionary to compile their
dictionary of collocations.

Finally, some remarks on the experimental evaluation:

- Despite the claim of generality of MinIE, the authors choose to experiment
only with ClausIE as underlying OIE system (most likely the optimal match). It
would have been very interesting to see if the improvement brought by MinIE is
consistent also with other OIE systems, in order to actually assess its
flexibility as a post-processing tool.

- Among the test datasets used in Section 7, I would have included the recent
OIE benchmark of Stanovsky and Dagan (2016), where results are reported also
for comparison systems not included in this paper (TextRunner, WOIE, KrakeN).

References:

- Manaal Faruqui and Shankar Kumar. Multilingual Open Relation Extraction using
Cross-lingual Projection. NAACL-HLT, 2015.

- Tobias Falke, Gabriel Stanovsky, Iryna Gurevych and Ido Dagan. Porting an
Open Information Extraction System from English to German. EMNLP 2016.

- Ndapandula Nakashole, Gerhard Weikum and Fabian Suchanek. PATTY: A Taxonomy
of Relational Patterns with Semantic Types. EMNLP 2012.

- Andrea Moro, Roberto Navigli. WiSeNet: Building a Wikipedia-based Semantic
Network with Ontologized Relations. CIKM 2012.

- Andrea Moro, Roberto Navigli. Integrating Syntactic and Semantic Analysis
into the Open Information Extraction Paradigm. IJCAI 2013.

- Claudio Delli Bovi, Luca Telesca and Roberto Navigli. Large-Scale Information
Extraction from Textual Definitions through Deep Syntactic and Semantic
Analysis. TACL vol. 3, 2015.

- Gabriel Stanovsky and Ido Dagan. Creating a Large Benchmark for Open
Information Extraction. EMNLP 2016.